# The Recovery of Soybean Plants after Short-Term Cadmium Stress

**DOI:** 10.3390/plants9060782

**Published:** 2020-06-22

**Authors:** Renata Holubek, Joanna Deckert, Inga Zinicovscaia, Nikita Yushin, Konstantin Vergel, Marina Frontasyeva, Alexander V. Sirotkin, Donald Samdumu Bajia, Jagna Chmielowska-Bąk

**Affiliations:** 1Department of Zoology and Anthropology, Faculty of Natural Sciences, Constantine the Philosopher University, ul. Nábrežie mládeže 91, 949-74 Nitra, Slovakia; renata.holubek@gmail.com (R.H.); asirotkin@ukf.sk (A.V.S.); 2Department of Plant Ecophysiology, Institute of Experimental Biology, Faculty of Biology, Adam Mickiewicz University, Poznań, ul. Uniwersytetu Poznańskiego 6, 61-614 Poznań, Poland; joanna.deckert@amu.edu.pl; 3Frank Laboratory of Neutron Physics, Joint Institute for Nuclear Research, 1419890 Dubna, Moscow Region, Russian; inga@jinr.ru (I.Z.); zing@nf.jinr.ru (N.Y.); vergel@jinr.ru (K.V.); marina@nf.jinr.ru (M.F.); 4Horia Hulubei National Institute for R&D in Physics and Nuclear Engineering, 30 Reactorului Str. MG-6, 077125 Bucharest–Magurele, Romania; 5Department of Biochemistry, Faculty of Science, The University of Bamenda, ENS Street, Bambili, Cameroon; donaldsamdumu.bajia@studenti.univr.it; 6Department of Biotechnology, University of Verona, Via San Francesco, 22, 37129 Verona VR, Italy

**Keywords:** metal stress, cadmium, cell viability, DNA methylation, lipid peroxidation, magnesium, manganese, potassium, recovery

## Abstract

Background: Cadmium is a non-essential heavy metal, which is toxic even in relatively low concentrations. Although the mechanisms of Cd toxicity are well documented, there is limited information concerning the recovery of plants after exposure to this metal. Methods: The present study describes the recovery of soybean plants treated for 48 h with Cd at two concentrations: 10 and 25 mg/L. In the frame of the study the growth, cell viability, level of membrane damage makers, mineral content, photosynthesis parameters, and global methylation level have been assessed directly after Cd treatment and/or after 7 days of growth in optimal conditions. Results: The results show that exposure to Cd leads to the development of toxicity symptoms such as growth inhibition, increased cell mortality, and membrane damage. After a recovery period of 7 days, the exposed plants showed no differences in relation to the control in all analyzed parameters, with an exception of a slight reduction in root length and changed content of potassium, magnesium, and manganese. Conclusions: The results indicate that soybean plants are able to efficiently recover even after relatively severe Cd stress. On the other hand, previous exposure to Cd stress modulated their mineral uptake.

## 1. Introduction

Cadmium is one of the most toxic metals posing serious concern to crop production and human health. There are several mechanisms of Cd toxicity. First, this metal leads to an over-production of reactive oxygen species (ROS). In plants, the ROS accumulation results from Cd-dependent disturbances in the electron transport chains in chloroplasts and mitochondria, depletion of the antioxidant system, and stimulation of the superoxide anion (O_2−_) generating, membrane-bond enzyme NADPH oxidase [1,2,3,4,5]. The resulting increase in ROS levels leads to oxidative stress and associated damage of lipids, proteins, and nucleic acids [6]. One of the most common symptoms of Cd toxicity in plants is the elevated level of oxidative stress markers such as thiobarbituric acid reactive substances (TBARS), including malondialdehyde (MDA), or carbonylated proteins [5,7,8,9,10,11]. Another mechanism of Cd toxicity is dependent on the ability of Cd to bind directly to biological molecules leading to changes in their conformation and therefore functioning. It is interesting that Cd can substitute for essential elements in molecules [12]. Some examples of this phenomenon of molecular mimicry are the substitution for Ca^2+^ in calmodulin molecules in radish, replacement of Mg^2+^ in chlorophyll in the aquatic plant *Ceratophyllum demersum*, or the exchange for Zn^2+^ in the *Arabidopsis* SUPERMAN (SUP37) transcription factor. In all of the described cases, the replacement of essential ions led to disturbances in the normal functioning of the affected molecules [13,14,15]. Finally, Cd stress in plants is frequently associated with alerted mineral homeostasis, which might at least partially, be the result of competition between Cd and other elements in membrane channels and/or transporters [8,16,17].

There are numerous studies that focused on Cd toxicity in plants [18,19,20,21]. However, information concerning the recovery process after exposure to this metal is very limited. It has been shown in *Lemna minor* that the post-stress recovery period differed among four applied metals, including Cd, Cu, Ni, and Zn. The longest recovery was observed after Zn treatment and the most rapid in the case of Ni. Cd showed the highest toxicity of the four applied metals. However, after the return of exposed plants to normal conditions, the plants rapidly recovered from Cd stress treatment, reaching 80% of relative growth after 4 days. The authors of the research pointed out to the importance of post-stress studies and suggested that recovery assays should be included in the ecological assessment tests [22]. In turn, Fojtová (2002) demonstrated that tobacco suspension cells exposed to Cd for 3 days were able to fully restore their growth after the removal of the metal. The recovery period was associated with intensive DNA repair. It was also reported that even relatively slight extension of Cd stress by one day led to the rapid decrease in cell viability and completely inhibited growth restoration of the cells. This result provides evidence that after a particular time of stress duration the recovery is no longer possible [23].

So far there is limited information concerning the recovery process after exposure to Cd in land plants grown in vivo. The present study is focused on the examination of the recovery process after relatively severe Cd stress in soybean, one of the most cultivated plant species worldwide. The physiological state of the seedlings, including growth, cell viability, lipid peroxidation, and photosynthesis parameters were evaluated directly after Cd stress and after 7 days of recovery period. Additionally, the mineral composition and photosynthesis parameters were assessed in the recovered plants. An interesting phenomenon associated with the exposure of plants to unfavorable conditions is stress memory, which is reflected by more rapid and/or robust reaction to repeated stress factors. This is most likely dependent on epigenetic modifications, such as changes in DNA methylation [24,25]. The present research includes evaluation of the global DNA methylation level directly after Cd exposure and after post-stress recovery.

## 2. Results

### 2.1. Cadmium Content

The Cd content in seedlings and recovered plants was dependent on the applied concentration (Figure 1). The metal content in relation to the one gram (1.0 g) of dry weight was much higher in the case of stressed seedlings than in the recovered plants (Figure 1). The main site of Cd accumulation in seedlings and recovered plants was in the roots. However, the distribution in organs differed depending on the applied concentration (Appendix A). Seedlings treated with lower Cd concentration accumulated 81% of the metal in the roots and 19% in the hypocotyls. The seedlings treated with higher Cd concentration contained 93% of metal in the roots and only 7% in the hypocotyls. Similar accumulation profile was observed in recovered plants. The plant previously exposed to Cd at the concentration 10 mg/L accumulated 62% of the metal in roots, 33% in the stems, and 5% in the first leaves. Plants stressed with Cd at the concentration 25 mg/L contained 85%, 11%, and 4% in the roots, stems, and in the first leaves, respectively. In stressed seedlings and recovered plants the translocation factor (TF), which estimates plant ability to translocate metals from the roots to the shoots, decreased under higher Cd concentration (Appendix A).

### 2.2. Growth Parameters

Exposure to Cd led to the inhibition of seedling growth and the appearance of brown stains on the roots (Figure 2A). The length of the roots was inhibited by nearly 40% in the seedlings exposed to lower Cd concentration and over 60% in the seedlings treated with the higher Cd concentration. Hypocotyls were less affected by the metal, however, a significant inhibition of their length has been observed in the case of seedlings stressed with the 25 mg/L Cd treatment (Figure 2B). Application of Cd at higher concentration also led to a significant decrease in the fresh weight of the roots (Figure 2C).

The morphology of the plants returned to normal growth conditions for 7 days post their exposure to Cd did not differ significantly from the control plants, with an exception of the roots shortening and browning noted in plants exposed earlier to the higher concentration of the metal (Figure 3A,B). The roots of these plants were reduced in length by 60% compared with the control. Previous exposure to Cd did not affect any other growth parameter of the recovered plants (Figure 3B,C).

### 2.3. Cell Death and Membrane Integrity

The roots of seedlings exposed to Cd at the higher concentration were characterized by an increase in Evans Blue uptake by over 150% in relation to the control, indicating a large increase in the number of dead cells (Figure 4A). Treatment with Cd at the higher concentration also led to statistically significant increase in TBARS level indicating enhanced lipid peroxidation (Figure 4B). There were no significant differences in cell mortality or the level of lipid peroxidation between control plants and recovered plants (Figure 4A,B).

### 2.4. Photosynthesis Parameters of Recovered Plants

Photosynthesis is one of the most Cd-sensitive processes. Therefore, chlorophyll content and the ratio between variable fluorescence and maximum fluorescence (Fv/Fm) has been examined in the plants post recovery period. The control and Cd-stressed plants showed no differences in the amount of chlorophyll *a* and *b* (Figure 5A) or Fv/Fm ratio (Figure 5B).

### 2.5. Content of Essential and Non-Essential Elements in Recovered Plants

The content of elements in various organs of recovered plants has been assessed using neutron activation analysis (NAA). In the case of macronutrients the content of three elements, potassium (K), calcium (Ca), and magnesium (Mg) has been measured. The distribution differed depending on the organ. The highest level of Mg was noted in roots, Ca was determined to accumulate in the stems, while the level of K was similar in all analyzed organs (Figure 6). Past exposure to Cd at the concentration 10 mg/L led to significantly higher content of K in the roots and stems of recovered plants (Figure 6A). Past treatment with Cd at the concentration of 25 mg/L resulted in the increase of Mg content in the leaves (Figure 6C).

The level of four micronutrients, chlorine (Cl), sodium (Na), manganese (Mn), and zinc (Zn), was also determined. Generally, the highest levels of Cl, Na, and Mn were detected in the roots, while Zn was evenly distributed among the organs (Figure 7). Past exposure to Cd at the highest concentration resulted in the increase of Mn content in plant roots and leaves (Figure 7C). Some non-essential elements, namely aluminum (Al), arsenic (As), bromine (Br), and rubidium (Rb), have been also detected in recovered plants. Their content was generally low and was unaffected by past Cd stress (Appendix A).

### 2.6. Examination of Global DNA Methylation Level

In order to evaluate the putative epigenetic changes induced by Cd, the level of 5′-methyl-2′-deoxycytidine (5MedCyd) has been assayed with the use of ELISA test. The level of global DNA methylation expressed as µM of 5MedCyd/1.0 µg of DNA was similar in the case of seedlings and recovered plants. No differences were observed between control plants and Cd-treated plants, neither directly after metal treatment nor after the recovery period (Figure 8).

## 3. Discussion

Contamination of the environment with Cd constitutes a serious problem in the world. High Cd levels noted in soil and sediments in various parts of the world are toxic to all living organism. In plants Cd-dependent alterations in homeostasis lead to hampered growth and decreased yield. Importantly, this metal is readily absorbed by crop plants and hereby enters the human food chain [26,27,28]. The adverse impact of Cd on plants is well documented [18,19,20,21]. However, the research concerning post stress period and plant recovery from metal stress is very scarce. The aim of the present study was to examine if soybean seedlings are able to efficiently recover from relatively severe Cd stress and to assess if past Cd exposure affected their future growth and metabolism. In order to answer the above mentioned questions, soybean seedlings were treated for 48 h with Cd solutions at concentration of 10 and 25 mg/L. Thereafter, they were transferred for 7 days to optimal growing conditions for the recovery period.

The study shows that Cd is efficiently absorbed by soybean seedlings and that its uptake is proportional to the applied metal concentration. Cd was also present in recovered plants (Figure 1A,B). The roots were the main site of Cd accumulation, although significant amount of the metal was transferred also to other organs (Appendix A). Cd was detected even in the first leaves of recovered plants. A significant finding was that, the metal translocation factor decreased with the applied concentration, suggesting that under severe Cd stress the root-to-shoot transport is somehow hampered (Appendix A).

Cd clearly exhibited toxic effect on soybean seedling. Even relatively short (48 h) exposure to the metal resulted in inhibition of growth (Figure 2A–C), increase in the amount of dead cells (Figure 4A), and increase in the abundance of lipid peroxidation products (Figure 4B). These are typical symptoms of Cd stress observed in numerous plant species [18,19,20,21]. A notable finding was that, after transfer to normal growth conditions the soybean plants efficiently restored their growth and managed to fully develop first leaves (Figure 3A–C). No differences in any of the growth parameters were noted in plants previously exposed to lower Cd concentrations (Figure 3A,B). In the case of plants treated with higher Cd concentration, the only noted change in their growth comprised reduction of the roots length (Figure 3A–C). Plants efficiently recovered also from other symptoms of Cd toxicity—the level of cell mortality and lipid peroxidation was found same in the control and in the recovered plants (Figure 4A,B).

Photosynthesis is one of the most Cd-sensitive processes. It has been demonstrated that even a concentration of 20 nM of Cd can inhibit the photosynthetic light reaction in the aquatic plant *Ceratophyllum demersum* [13]. Here we show that the first leaves of recovered plants contained around 10 µg of Cd per 1.0 g of dry weight but no disturbances in photosynthesis have been observed. The chlorophyll content and Fm/Fv ration did not differ significantly from the same parameters in the control plants (Figure 5). Taken together, the results obtained demonstrate that although short term Cd stress led to the development of various symptoms of toxicity, after transfer to normal growth conditions the soybean plants efficiently restored their growth and metabolism.

So far there is little information concerning the capacity of plants to recover from metal stress. Study on common duckweed (*Lemna minor*) showed that it can restore growth to some extent after exposure to Cd, Zn, Cu, and Ni [22]. In *Tetradenia riparia* exposure to Cd resulted in growth inhibition and attenuated CO_2_ assimilation. After a 3-week recovery period these parameters improve to some extent, although they did not reach the levels observed in the control [29]. In tobacco suspension cells the threshold for recovery from Cd stress has been estimated at 3 days. Prolongation of this period by only one day resulted in the death of all cells. It was suggested that the recovery process is dependent at least partially on the efficiency of the DNA repair mechanisms. Exposure to the metal resulted in damage of genetic material, while the recovery period was accompanied by DNA repair [23]. Rice plants exposed to Al for a relatively short time (24 h) were characterized by a decrease in the cells mitotic index and several chromosomal abnormalities. These symptoms of metal toxicity were observed even after 15 days of recovery period. The abnormalities in chromosomes included chromosome stickiness, lagging, fragmentation, and formation of bridges. In addition, around 65% of pollen in mature rice plants recovered from previous Al stress were sterile in comparison to only 10% pollen sterility observed in the control [30]. Treatment of *Acacia raddiana, Acacia tortilis,* and *Prosopis juliflora* seeds with Cu or Pb resulted in decrease in germination rate. After withdrawal from exposure to the metals *P. juliflora* seeds restored the germination capacity. On the other hand, *A*. *raddiana* and *A*. *tortilis* seeds still maintained low germination rate [31]. Studies on green algae *Scenedesmus* sp. showed that treatment with Cu and Zn resulted in significant decrease of nitrate reductase (NR) activity and nitrate uptake. After withdrawal from metal stress the alga cells were able to fully restore NR activity within 96 h [32].

Interestingly, although excess of metals is undoubtedly toxic, there is increasing evidence that even non-essential metals might also exhibit beneficial effects. Several studies showed that pre-treatment with metals stimulates defense mechanisms and induces tolerance toward other stress factors such as osmotic stress, salinity, drought, low temperature, and pathogen attack. Additionally, exposure to low concentrations of some trace elements, including Cd, results in stimulation of growth and increase in pigments level [33,34,35]. Metal treatment can also enhance uptake of other elements. For instance, Cd-dependent increase in Fe accumulation has been noted in the roots, stems, and leaves of hydroponically grown cucumber and in *Potamogeton crispus* leaves [36,37]. In the latter case leaves of Cd-treated plants contained also higher levels of Ca, Mn, and Na [37]. A comprehensive study of the mineral composition in *Pfaffia glomerata* hyperaccumulator revealed that Cd treatment resulted in elevated levels of K in the leaves and Cu in the roots [38]. Tomato cultivars exposed to this metal accumulated higher amounts of B in the roots, stems, and leaves and of Mn in the stems and leaves [39]. Present study evidences that Cd-dependent increase in the uptake of minerals lasts even after the cessation of metal treatment. The obtained results demonstrate that plants recovered from Cd stress contained higher levels of K in the roots and stems, Mn in the roots and Mg in the leaves when compared to the control (Figure 6A,C and Figure 7C). Observed accumulation could occur only during the recovery periods, when the plant was grown in the soil. The results indicate that post Cd exposure promotes the future uptake of essential minerals. One of the possible explanations of this phenomenon is Cd-dependent induction of the expression of particular transporters or chelating compounds. Indeed, it has been demonstrated in the W6nk2 barley cultivar that exposure to Cd leads to increased expression of numerous transport related genes including ZIP-like zinc transporters, putative ABC transporters, putative potassium transporters, putative phosphor translocators, and putative ammonium transporters [40]. Similarly, genes encoding zinc, copper, sulphate, ammonium, and nitrate transporters showed elevated expression in response to Cd in tomato plants, while date palm exposure to this metal resulted in higher transcript levels of the *Pdabcc*, *Pdhma2*, *Pdmate5*, and *PdNramp6* transporters genes [41,42,43]. Additionally, Cd induces biosynthesis of metal phytochelatines, which act as metal chelators [44]. It can be thus assumed that Cd-dependent increase in the expression of metal transporters and chelators leads to enhanced capability of mineral uptake by the plants, even after the decline of metal stress.

There is increasing evidence that plants possess a specific memory, which enables them more robust response to repeated stress factors. For instance, plants subjected to recurrent drought or pathogen attack show less pronounced symptoms of damage. This phenomenon is most likely dependent on epigenetic changes, which include histone modifications and changes in DNA methylation pattern, leading to the rearrangement of chromatin. Increased accessibility of particular chromatin regions to regulatory proteins enables faster transcription of stress-related genes and more rapid response to unfavorable conditions [24,25]. To evaluate if past Cd stress left a trace on the epigenetic level, the amount of the DNA methylation marker, 5′-methyl-2′-deoxycytidine (5MedCyd) has been measured. No differences between the control and Cd-treated plants have been observed neither directly after metal exposure nor after the recovery period (Figure 8). Although the results indicate that there is no change in global DNA methylation level, this did not rule out possible epigenetic changes in response to Cd. Methylation and demethylation events could occur on a smaller scale in particular DNA regions without affecting the global level. Indeed, a comprehensive study on rice showed that although Cd does not affect global methylation of cytosine in DNA, it modifies methylation pattern of specific genes [45]. Similarly, it has been show that exposure to Cd modulates methylation of *osZIP1* gene in rice and promotor regions in wheat transporter genes [46,47]. Second, epigenetic changes could be driven by other mechanisms for example chemical modification of histones. In *Tetradenia riparia,* Cd treatment results in the decline in lysine methylation of histone H3 [29]. This metal was also found to affect the expression of several genes encoding histone deacetylases (HADCs) in cotton, which indicates Cd-dependent modulation of histones acetylation [48]. In turn exposure of soybean cell suspension to Cd resulted in increased expression of histone H2B [49].

In summary, the present study shows that soybean seedlings are capable of efficient recovery even after severe Cd stress. Additionally, the recovered plants exhibit higher capacity for uptake of some essential minerals, namely K, Mn, and Mg. So far there is limited number of studies on plants recovery from metal stress. One of the possible reasons is that metals are stable elements and therefore metal stress is not as transient as other unfavorable conditions such as low and high temperature or drought. However, the level of metals in the soil can still significantly decrease due to washing out or uptake by plants and other organisms [33]. In turn, assessment of recovery period is crucial for precise understanding of plant tolerance mechanisms, which has been elegantly demonstrated by Striker (2011) in an example which involved the effects of flooding [50]. The author describes two strategies adopted by plants during unfavorable conditions. During flooding, plants can either keep growing using mainly carbohydrates reservoirs (type A strategy) or suspend the growth and save energy for the future (type B strategy). After cessation of stress period, the type A plants, depleted from carbohydrates, exhibit low growth rate. However, type B plants restore their metabolism and grow much faster, sometimes even faster than in the control conditions. Therefore, the common practice of assessing stress intensity by measurements of plants growth straight after stress conditions can be misleading. With this approach plants exhibiting type A strategy would be considered as stress resistant, while plant showing type B strategy—stress sensitive. Inclusion of recovery studies would show that actually the type B strategy might be more favorable for stress endurance. Specially designed studies on recovered plants can also give valuable information on the phenomena of plants memory and stress cross-tolerance.

## 4. Materials and Methods

### 4.1. Plant Cultivation and Treatment

The cultivation and treatment procedures were carried out as described earlier [51]. Soybean seeds, kindly supplied by Department of Genetics and Plant Breeding at the Poznań University of Life Sciences, were surface sterilized with 75% ethanol for 5 min and afterwards with 1% sodium hypochlorite for 10 min. Seeds were washed for 30 min under running water and soaked in distilled water for 2–3 h. Germination process took place at room temperature during 48 h on plastic trays with two layer of lignin and one layer of moistened blotting paper covered with aluminum foil. Germinated seedlings selected with respect to similar roots length, were transferred to new Petri dishes (10 cm of diameter), wherein the roots were placed between two layers of blotting paper in cut-out holes. Thereafter, they were treated for 48 h with 5 ml of distilled water (control) and 5 ml of CdCl_2_ solution in two Cd concentrations: 10 mg/L (89 µM) and 25 mg/L (223 µM). The same concentrations were used in earlier studies performed at the Department of Plant Ecophysiology, Adam Mickiewicz University in Poznań and were initially chosen on the basis of the root tolerance index [51,52,53]. For recovery experiments control seedlings and seedlings stressed with Cd for 48 h were transferred to plastic pots of 18 cm of diameter with sterilized commercial, universal soil (pH 5.5–6.5, salinity at max. 1.9 g NaCl/dm^3^). They were grown for 7 days in a growing chamber, with temperature 22 °C and photoperiods 12/12. Plants were watered every day with tap water.

### 4.2. Measurement of Cd Content

The amount of Cd in the studied samples was determined by iCE 3300 AAS Atomic Absortion Spectrometer with electrothermal (graphite furnace) atomization (Thermo Fisher Scientiic, Waltham, MA, USA). AAS-Cd standard solution (Merck, Darmstadt, Germany) with concentration of 1.0 g/L was used for preparation of standard solutions. The samples were placed in a Teflon vessel and treated with 3 mL of concentrated nitric acid (HNO_3_) and 1.0 mL of hydrogen peroxide (H_2_O_2_). For full digestion the samples were put into the microwave digestion system (Mars; CEM, Matthews, NC, USA). Digestion was performed in two steps: (1) ramp: temperature 160 °C, time 15 min, power 400 W, and pressure 20 bar; (2) hold: temperature 160 °C, hold time 10 min, power 400 W, and pressure 20 bar. The digests were quantitatively transferred to 100 mL calibrated flasks and made up to the volume with bi-distilled water. The quality control of AAS was ensured using NIST standard certified reference materials (NIST, Gaithersburg, MD, USA): 1570a (Trace Elements in Spinach Leaves) and 1575a (Trace Elements in Pine Needles). Measurements were performed on samples from three independent experiments.

### 4.3. Measurements of Growth Parameters

Growth parameters were measured after 48 h of stress in the case of seedlings and after 7 days of the growth in optimal conditions in the case of recovered plants. For fresh weight the roots, hypocotyls, and in the case of recovered plants the epicotyls, were cut off on ice and weighted immediately on the laboratory scale. Dry weight was measured after 72 h long incubation in 55 °C. Measurements were performed on samples from three independent experiments, each consisting of a pool of 20 seedlings or 10 recovered plants.

### 4.4. Estimation of Cell Viability

Cell viability was estimated on the basis of Evans Blue uptake according to Lehotai et al. 2011 [54]. Approximately 200 mg of roots were cut off on ice and incubated for 20 min in 0.25% Evans Blue (Sigma-Aldrich, St. Louis, MO, USA, E-2129). Then they were washed twice for 15 minutes in distilled water and homogenized using a mortar and pestle, containing destaining solution consisting of 50 mL of ethanol, 49 mL of distilled water and 1.0 mL of 10% SDS. Samples were incubated in heating block for 15 min at 50 °C and centrifuged (12,000 rpm, 20 °C, 15 min). The Evans Blue uptake, indicating cells death, was measured spectrophotometrically at λ = 600 nm. De-staining solution was used as the blank. Measurements were performed on samples from three independent experiments, each consisting of a pool of 10 seedlings or 3 recovered plants.

### 4.5. Evaluation of Membrane Integrity

Membrane damage has been assessed indirectly by measurement of lipid peroxidation evaluated on the basis of the amount of thiobarbituric reactive substances (TBARS) according to Cuypers et al. (2011) with minor modifications [55]. Approximately 200 mg of roots were cut off on ice and homogenized with 3.0 mL of 10% TCA buffer (Sigma-Aldrich, Sigma-Aldrich, St. Louis, MO, USA TO699). After centrifugation (12,000 rpm, 4 °C, 10 min) 1.0 mL of supernatant was transferred to glass tubes, filled with 4.0 mL of 0.5% TBA (Sigma, Sigma-Aldrich, St. Louis, MO, USA T5500) dissolved in 10% trichloroacetic acid (TCA) and incubated for 30 min in 95 °C. After heating the samples were cooled, mixed by inversion and centrifuged (5000 rpm, 4 °C, 2 min). The absorbance of the supernatant was measured at λ = 532 nm and corrected for unspecific absorbance at λ = 600 nm. Measurements were performed on samples from four independent experiments, each consisting of a pool of 10 seedlings or 3 recovered plants.

### 4.6. Measurements of Chlorophyll Content and Photosynthesis Parameters

For measurements of chlorophyll content, leaves of recovered plants were cut off on ice (200 mg) and incubated in 7.0 mL dimethyl sulfoxide (DMSO) at 65 °C for 2 h. Thereafter they were cooled to room temperature, centrifuged, and the absorbance of supernatant was measured at λ = 665 nm and λ = 649 nm. DMSO was used as the blank. Chlorophyll content was calculated using the following formulas according to Wellburn 1994 [56]:
a = 12.9*(A665) − 3.45*(A649) and chlorophyll b = 21.99*(A649) − 5.32*(A665)(1)

For estimation of photosynthesis efficiency, the plants were kept in a dark room for 30 min and afterwards the ratio of variable fluorescence and maximum fluorescence (Fv/Fm) was measured with the use of chlorophyll fluorimeter (FluorPen FP100; Photon Systems Instruments, Drásov, Czech Republic). Results of chlorophyll content and photosynthesis parameters are presented as the mean of three independent experiments, each consisting of 10 plants.

### 4.7. Assessment of the Elements Content

Elemental composition of the samples was determined by neutron activation analysis at the IBR-2 reactor, Dubna, Russia. Irradiation was carried out under two experimental conditions; short irradiations of 3 min at a thermal neutron flux of approximately 1.6 × 10^13^
*n* cm^−2^ s^–1^ for the determination of Al, Ca, Cl, Cu, Mg, and Mn. Long irradiation of 3 days at a neutron flux of 1.8 × 10^11^ cm^−2^ s^−1^ was used to determine the main part of elements: Na, K, Sc, Fe, Co, Ni, Zn, As, Br, Rb, Sr, and Cs.

After the appropriate decay times, gamma spectra of induced activity were obtained using three spectrometers based on HPGe detectors with an efficiency of 40–55% and resolution of 1.8–2.0 keV for total-absorption peak 1332 keV of the isotope 60Co and Canberra spectrometric electronics.

The analysis of the spectra was performed using the Genie2000 software from Canberra, with the verification of the peak fit in an interactive mode, while the calculation of concentration was carried out using software “Concentration” developed in FLNP [57].

The quality control of the analytical measurements was carried out using certified reference materials: NIST SRM 1573—Tomato leaves, NIST SRM 1573—Peach leaves, NIST SRM 1633c—Coal fly ash, and NIST SRM 2710—Montana Soil. Reference materials were analyzed under the same experimental conditions as the samples. The obtained values for concentrations of standard reference materials differed from the certified values in the range (0.2–7%). Measurements were performed on samples from three independent experiments.

### 4.8. Analysis of the Level of DNA Methylation

The level of DNA methylation was assayed on the basis of Global DNA Methylation ELISA Kit (5′-methyl-2′-deoxycytidine Quantitation) (STA-380, Cell Biolabs, inc., San Diego, CA, USA). Roots and/or leaves of seedlings were cut off on ice, frozen in liquid nitrogen, and stored in −80 °C. The DNA was isolated using Clean Plant DNA kit (CP-D00096; GC biotech B.V., Waddinxveen, The Netherlands) according to manufacturer’s instructions. The concentration and purity of DNA were measured on NanoCell (Thermo Scientific, Waltman, MA, USA) at spectrophotometer Biomate^TM^ 3S (Thermo Scientific, Waltman, MA, USA). For ELISA kit 5.0 µg of DNA from each experimental variant was denaturated for 5 min at 95 °C following 2 h long digestion by 20 Units of nuclease S1 (Bio Shop Canada, Burlington, ON, Canada, NUC333.50) and 1 h long digestion by 10 Units of alkaline phosphatase (Sigma-Aldrich, Sigma-Aldrich, St. Louis, MO, USA P6774-2KU). The digestion was carried out at 37 °C. Further procedures were performed according to the manufacturer’s instructions. The absorbance of the samples was measured on İMARK^TM^ Microplate Reader (Bio-Rad, Hercules, CA, USA) and the 5′-methyl-2′-deoxycytidine (5MedCyd) concentrations were calculated using İMARK^TM^ Microplate Reader software (Bio-Rad, Hercules, CA, USA). The measurements were performed in two technical repetitions, on samples from three independent experiments.

### 4.9. Statistical Analysis

The significant differences in relation to the control were calculated with the use of ANOVA by *p* < 0.05 using Free Statistics Calculators version 4.0.

## 5. Conclusions

Despite the importance of recovery studies so far the topic concerning plant regeneration after exposure to metals is largely overlooked. The present study complexly examines the recovery of soybean plants after Cd stress. The results show that initial metal stress led to the development of toxicity symptoms including growth inhibition, increased cell death, and membranes damage. However, after transfer to optimal conditions, the plants were able to efficiently recover and restore their growth. On the other hand, the recovered plants were characterized by higher content of K, Mg, and Mn. These results indicate that past exposure to Cd modulated future mineral metabolism of plants. Neither direct nor past Cd exposure had a significant effect on the level of one of the epigenetic markers—global DNA methylation. However, further studies would be needed to rule out with certainty the possible impact of Cd stress on epigenetic modifications and stress memory.

## Figures and Tables

**Figure 1 plants-09-00782-f001:**
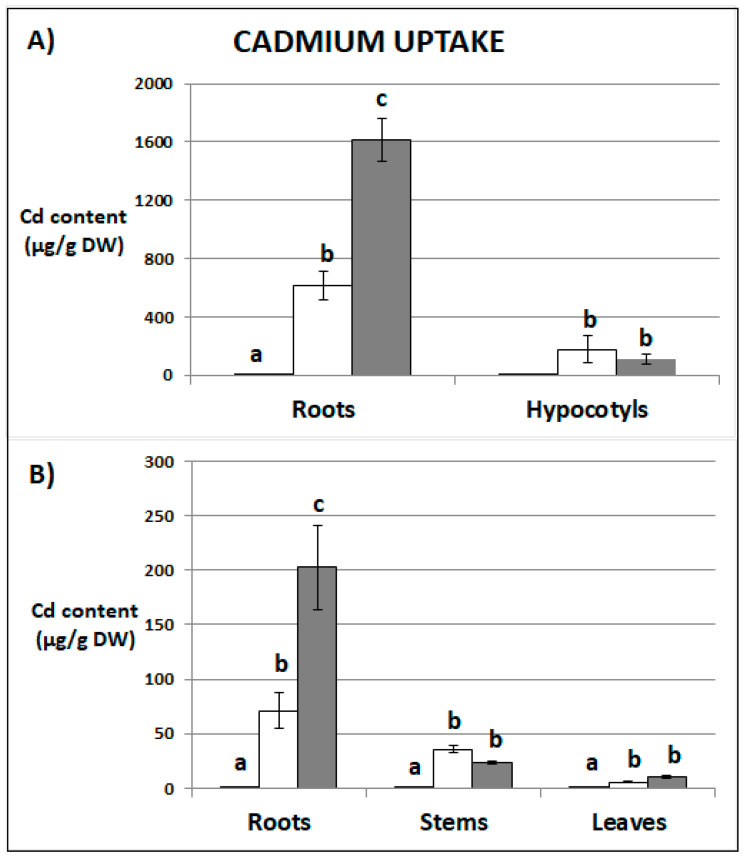
Cadmium content in roots and hypocotyls of soybean seedlings (**A**) and roots, stems, and first leaves of recovered plants (**B**). Black bars—control plants, white bars—plants stressed previously with Cd at the concentration 10 mg/L, dark grey bars—plants stressed previously with Cd at the concentration 25 mg/L. Results are the means of three independent experiments ± *SE*. Results showing no statistically significant differences in relation to control by *p* = 0.05 are marked with same letter.

**Figure 2 plants-09-00782-f002:**
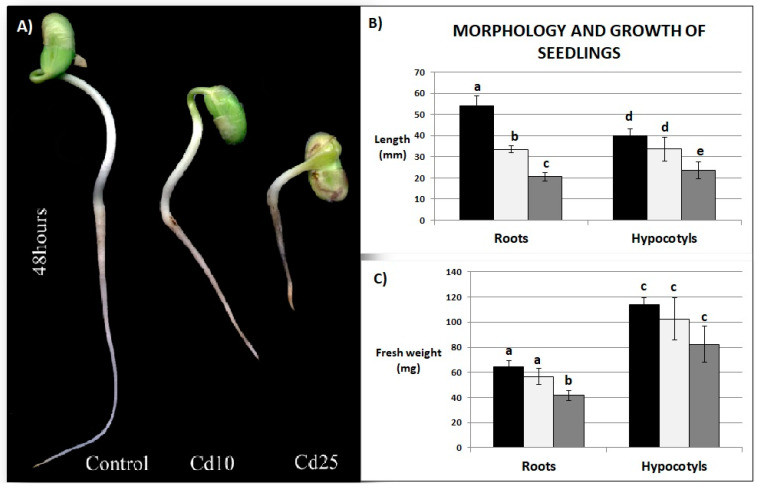
Impact of Cd on seedlings morphology (**A**), roots and hypocotyls length (**B**) and fresh weight (**C**) Black bars—control plants, white bars—plants stressed previously with Cd at the concentration 10 mg/L, dark grey bars—plants stressed previously with Cd at the concentration 25 mg/L. Results are the means of three independent experiments ±*SE*. Results showing no statistically significant differences in relation to control by *p* = 0.05 are marked with same letter.

**Figure 3 plants-09-00782-f003:**
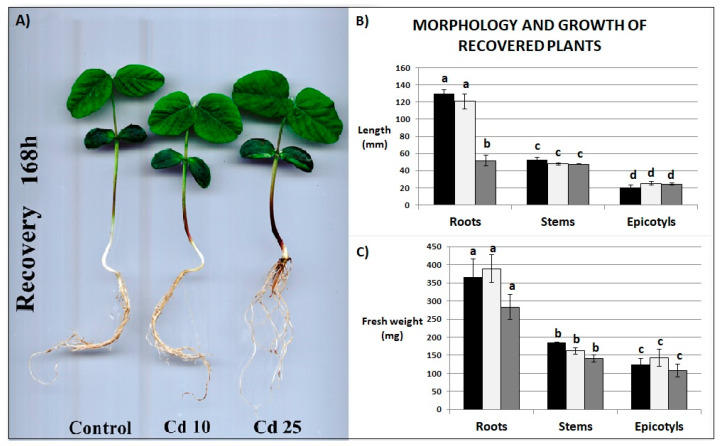
Morphology of recovered plants (**A**), length (**B**) and fresh weight (**C**) of roots, stems and epicotyls. Black bars—control plants, light grey bars—plants stressed previously with Cd at the concentration 10 mg/L, dark grey bars—plants stressed previously with Cd at the concentration 25 mg/L. Results are the means of three independent experiments ±*SE*. Results showing no statistically significant differences in relation to control by *p* = 0.05 are marked with same letter.

**Figure 4 plants-09-00782-f004:**
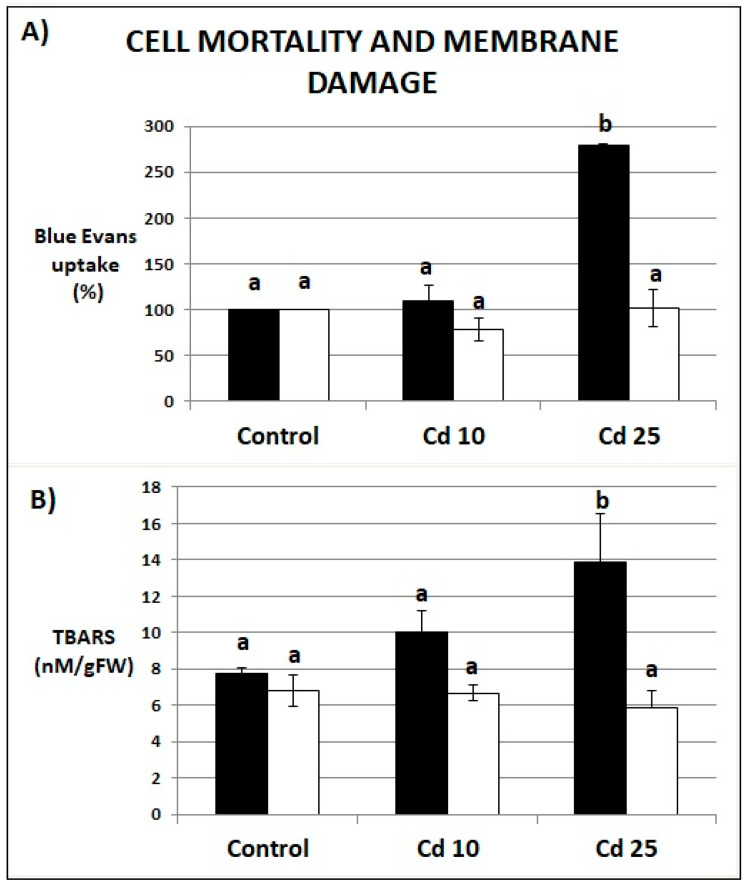
Evans Blue uptake indicating cell death expressed as percentage in relation to the control (**A**) and lipid peroxidation expressed as increase in thiobarbituric acid reactive substances (TBARS) level (**B**) in seedlings after 48 h of Cd stress (black bars) and after 7 days of recovery period (white bars). Results are the means of 3–4 independent experiments ±*SE*. Results showing no statistically significant differences in relation to control by *p* = 0.05 are marked with the same letter.

**Figure 5 plants-09-00782-f005:**
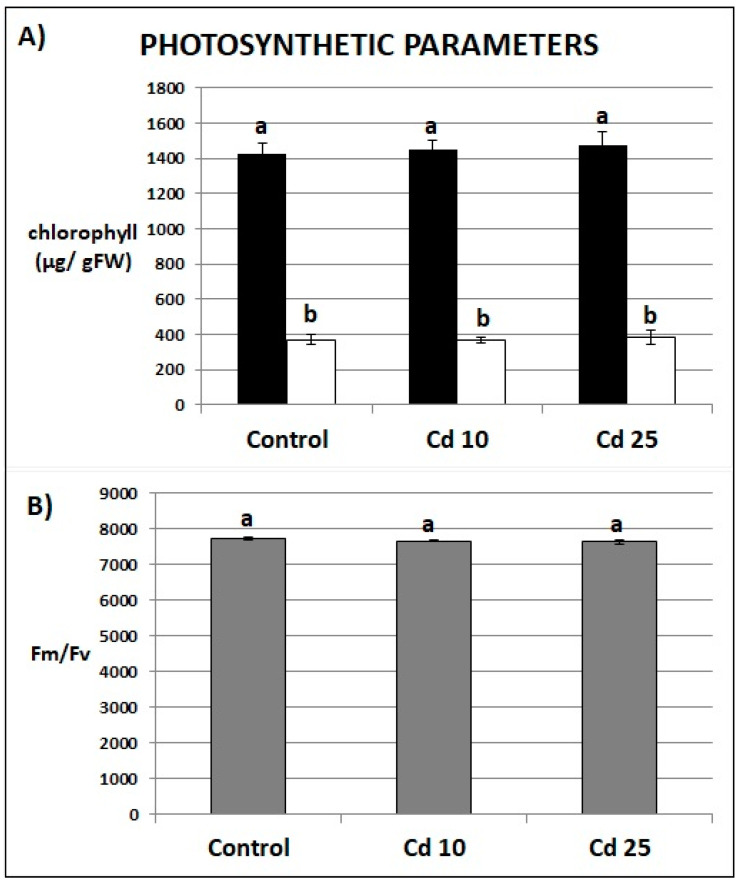
Content of chlorophyll *a* (black bars) and chlorophyll *b* (white bars) (**A**) and Fv/Fm ratio (**B**) of plants after 7 days of recovery periods. Results are the means of three independent experiments ±*SE*. Results showing no statistically significant differences by *p* = 0.05 are marked with the same letter.

**Figure 6 plants-09-00782-f006:**
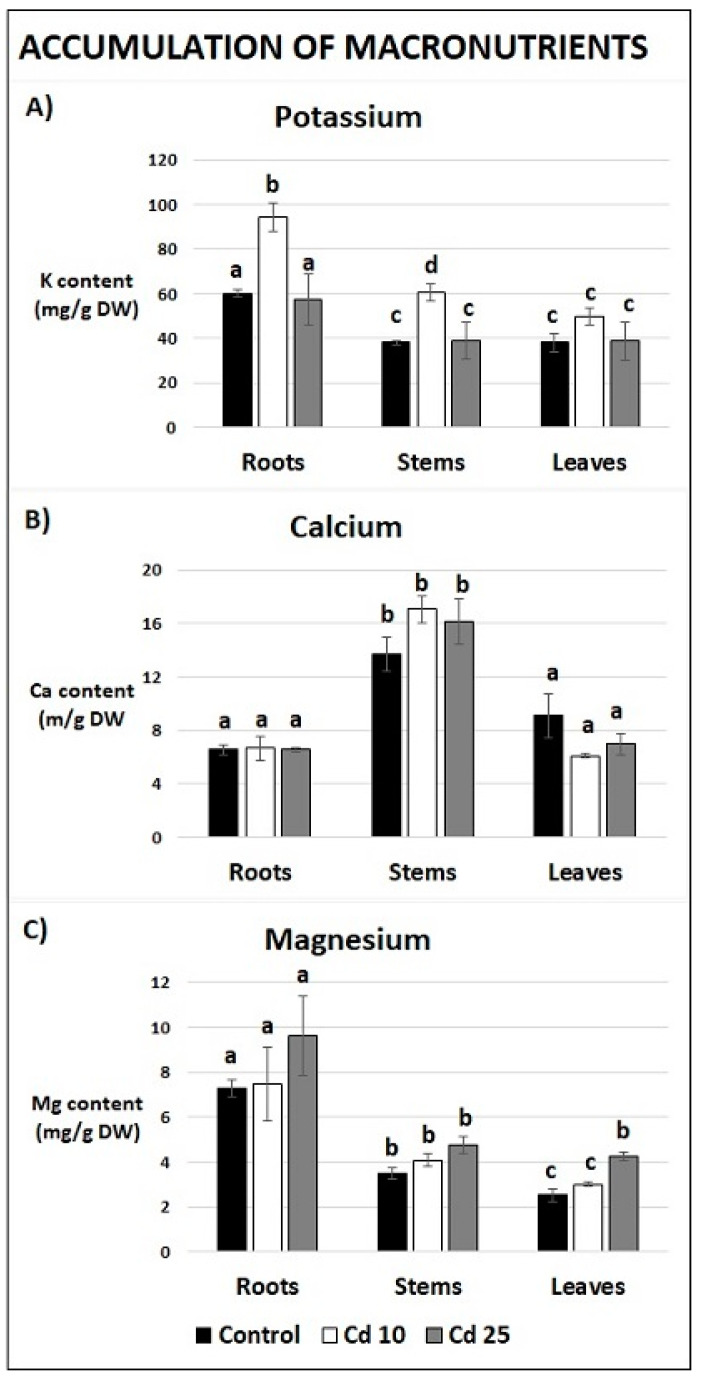
Content of macronutrients, potassium (**A**), calcium (**B**) and magnesium (**C**) in plants after 7 days of recovery. Results are the means of three independent experiments ± SE. Results showing no statistically significant differences by *p* = 0.05 are marked with the same letter.

**Figure 7 plants-09-00782-f007:**
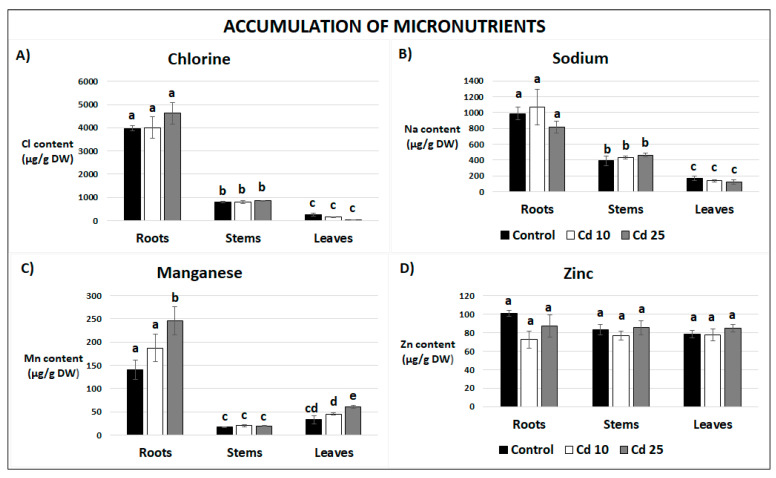
Content of micronutrients, chlorine (**A**), sodium (**B**), manganese (**C**) and zinc (**D**) in plants after 7 days of recovery periods. Results are the means of three independent experiments ± SE. Results showing no statistically significant differences by *p* = 0.05 are marked with the same letter.

**Figure 8 plants-09-00782-f008:**
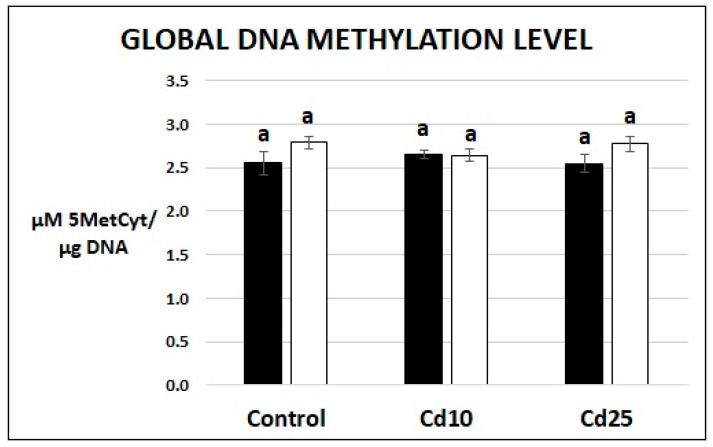
The level of global methylation in seedlings after 48 h of Cd stress (black bars) and after 7 days of recovery period (white bars). Results are the means of three independent experiments ± *SE*. Results showing no statistically significant differences in relation to control by *p* = 0.05 are marked with the same letter.

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
