# Peer review of "The Recovery of Soybean Plants after Short-Term Cadmium Stress"

_plants, 2020, doi:10.3390/plants9060782_

Round 1

Reviewer 1 Report

Although, from my point of view, the manuscript has improved after the authors' revision, the manuscript still lacks complete discussion. The authors have included some new references, however a critical discussion of the results compared to the most recent literature is missing, as already requested in the first review round. Based on this, in my opinion, the authors should further improve the discussion and emphasize the new findings of the work.

Author Response

Dear Reviewer,

We would like to thank you for all valuable comments.

As suggested the discussion has been in large part re-written. The changes (marked in the manuscript in blue colour) included:

  1. Reorganization of the discussion. The sequence has been changes to presentation and discussion of results concerning - cadmium toxicity, efficient post-stress recovery, observed increase in accumulation of nutrients in the recovered plants and post-stress epigenetic changes. The final paragraph highlights the importance of the studies on post-metal stress recovery.

  1. The part concerning plants recovery from metal stress has been extended on the basis of more comprehensive literature including studies on Cd and other metals. As the literature data on the topic is rather limited we believe that we have included most of the available references.

  1. The results showing increased accumulation of nutrients in the recovered plans have been discussed more detailly on the basis of more comprehensive literature.

  1. The paragraph describing possible epigenetic changes has been extended by literature-derived information on histone modifications.

  1. The last part of the discussion has been re-written to further highlight the importance of studies concerning post-metal stress recovery.

Additionally the manuscript has been send for professional language correction.

Reviewer 2 Report

Overall, the authors followed the suggested indications.

As minor points:

L96 change “decrease” into “decreased”

L117 and later, I suggest to change “past” into “previous”

L237-238 I suggest to change into “Noteworthy, in the case of tobacco cells full recovery was possible only after stress proceeded for maximum 3 days”

L258 change “adapted” into “adopted”

Author Response

Dear Reviewer,

We would like to thank for all valuable comments.

Please find below detailed reply to all comments and suggestions:

Comments and Suggestions for Authors

Overall, the authors followed the suggested indications.

As minor points:

L96 change “decrease” into “decreased”

The error has been corrected.

L117 and later, I suggest to change “past” into “previous”

The term has been changed.

L237-238 I suggest to change into “Noteworthy, in the case of tobacco cells full recovery was possible only after stress proceeded for maximum 3 days”

In the time course of the revision the whole fragment of the text has been changed.

L258 change “adapted” into “adopted”

The term has been changed.

Additionally in frame of the revision we have extended some parts of the discussion and unified formatting of the figures. The manuscript has been send for professional language correction.

All changes are marked in the manuscript in blue colour.

Reviewer 3 Report

Dear authors,

Please find my annotated PDF of your submitted manuscript. I have identified as many English language issues as I can, however, many remain and require amendment.

The text throughout the manuscript requires substantial English language editing - please have a native English speaking colleague proof your manuscript before its resubmission, or pay for a professional editing service for this purpose.

The Figures contain interesting and informative information, however, the quality of the presentation of some Figures is poor and require attention. You also need to be more consist with formatting from Figure to Figure so that each of the presented Figures appear related to each other and flow from one figure to the next.

If the material is available, then I would suggest you incorporate assessment of chlorophyll a and b content immediately post the Cd stress treatment in order to accurately assess the degree to which Cd stress inhibits photosynthesis.

The results and clearly discussed in the text of the Results section, and the interpretation of the data presented in the manuscript is well described in the text of the Discussion section of the manuscript. However, the impact of both of these sections would be significantly improved if they were not so heavily littered with English language issues. Therefore, please have the manuscript reviewed in order to address this issue.

Regards,

Andy.

Author Response

Dear Reviewer,

Thank you for all the correction – they have been included in the manuscript. Additionally, after modification of the discussion section the manuscript has been corrected by native English speaker colleague biologist and by professional editing service (Scribendi: https://www.scribendi.com).

In relation to some question included in the PDF file:

Line 87: We used the term “site” as we did not detect the cadmium in particular tissues but rather in the whole organs.

Line 192: The Cd content in the recovered plants has been assessed in the first leaves.

Line 193: The reference to the Figure (Supplementary Table 1) has been added in the text.

The Figures contain interesting and informative information, however, the quality of the presentation of some Figures is poor and require attention. You also need to be more consist with formatting from Figure to Figure so that each of the presented Figures appear related to each other and flow from one figure to the next.

The figures have been re-formatted: the headings have been included in all of the figures, the font size, charts size and colours have been unified, the statistical differences have been marked with letter in all the charts. Additionally to make the figures style more consistent presentation of the data on Figure 1 and Figure 2 has been changed.

If the material is available, then I would suggest you incorporate assessment of chlorophyll a and b content immediately post the Cd stress treatment in order to accurately assess the degree to which Cd stress inhibits photosynthesis.

Indeed, it would be more consistent if the chlorophyll a and b and photosynthesis parameters would be measured straight after metals stress and after the recovery periods. However, as the seedlings stressed with Cd had no developed leaves we do not suspect much change in the parameters. Unfortunately we do not have the material left.

The results and clearly discussed in the text of the Results section, and the interpretation of the data presented in the manuscript is well described in the text of the Discussion section of the manuscript. However, the impact of both of these sections would be significantly improved if they were not so heavily littered with English language issues. Therefore, please have the manuscript reviewed in order to address this issue.

We have further improved and extended some part of the discussion and thereafter submitted the manuscript for professional language correction.

Round 2

Reviewer 3 Report

Dear authors,

The revised version of your manuscript is much improved on the original version of your manuscript.

A few minor spelling and/or editing issues remain. These have been identified in the attached and annotated PDF of your revised manuscript.

One these changes have been addressed by the authorship team, the manuscript can proceed to the next stage of the publication process.

Regards,

Author Response

Dear Reviewer,

We would like to thank you for all valuable corrections.

We have included them in the revised manuscript and marked with blue colour.

with best regards

Jagna Chmielowska-BÄ…k

This manuscript is a resubmission of an earlier submission. The following is a list of the peer review reports and author responses from that submission.

Round 1

Reviewer 1 Report

The authors present a work entitled “The recovery of soybean plants after short term cadmium stress”. The topic is certainly of interest and to date very few works are dedicated to the description of the recovery of plants post-stress and in particular after cadmium stress. The manuscript is well written and organized, the experiments are adequately described and the data well presented. In my opinion, the work is worthy of being published on Plants. in order to improve the general quality of the work, I suggest only a few changes:

Ln 55: “There are numerous studies focused on Cd toxicity in plants”, please, add some references.

Ln 312: please, indicate the volume of DMSO

Ln 367: As a suggestion, in the conclusions, the authors could envisage a similar study using the same plants stressed by other heavy metals, so that this work can be part of a larger project.

Reviewer 2 Report

This paper reports a toxicity experiment using soy bean seedlings exposed to cadmium chloride dissolved in distilled water, then transplanted in soil and observed over 7 days. A good deal of measurements were conducted on these plants which all seem well-thought and follow appropriate methodologies. It is shown that the plants, although suffering from toxicity in the water solution, are able to restore normal growth after being transplanted. There are some longer-lasting changes in mineral content, especially potassium.

The authors are correct in that the issue of recovery after a toxic insult is underrated in environmental toxicology. However, it is unclear how recovery from heavy metal exposure would happen in the field. If a soil is contaminated with cadmium, it will be contaminated for a very long time, due to strong binding of metals to the soil matrix. Experiments simulating this use low chronic exposures, for the complete growing season. I don’t see the ecological conditions under which the exposure in this paper would occur (short-term exposure to a very high concentration in water, followed by zero exposure in soil). What kind of ecology is simulated by this?

That cadmium is toxic we know; we don’t need to be convinced of that. That recovery of plants from toxicity is a relevant issue, I can understand. But I would argue that this is relevant for chemicals with limited presence in the environment, chemicals with short half-lives that degrade in reasonable time, not with heavy metals which by definition cannot degrade and stay in the soil for a very long time.

Longer recovery studies, under conditions that represent some environmental scenario, including reproductive output of the mature plant, are needed. The present paper represents too small an advance.

Reviewer 3 Report

The Ms plants-780936 deals with the recovery of soybean seedlings from cadmium stress. The topic is surely interesting, and I agree with the Authors about the usefulness of recovery studies to have a more complete picture of plant tolerance, but there are some major issues to be solved.

The biggest problem is a physiological one: I think that any recovery study needs to be done on plants at the same physiological phase while results obtained on plants clearly on different stages could be misleading. In fact, differences of plant behavior could also depend on the specific plant development state. In addition, stress and recovered plants are grown on different growth medium as specified in the section MM.   In my opinion a “negative control”, with plants treated with cadmium without any recovery, should also be done.

In particular:

L33 I suggest to put keywords in alphabetical order

L45 actually TBARS include MDA

L65 please, specify and quantify the term “slight”

L71-72 I suggest to change into “….cell viability, lipid peroxidation and photosynthesis……”

L82 I suppose that the Authors mean ”stressed seedlings”

L91 is “with” for “in”?

L102 I suggest to change into “….seedlings under 25 mgL-1 Cd treatment”

L164 in table 2S significance letters should be added

L182 the Authors could calculate also TF (translocation factor)

L213 DNA methylation is  surely an interesting parameter, but as it stands, does not add much information. Could the Authors analyze more in detail this parameter?

L214 the Authors should  express the molarity of the Cd concentrations used in their work to make possible the comparison with previous studies. This applies not only to photosynthesis but also to other parameters, when a comparison is made with literature

L264 I suppose that “whole” stands for “hole”

L265 specify also at this point the days of cadmium treatment

L301 the Authors should underline that this method gives an indirect measure of membrane damage